# Experimental Studies of the Liposomal Form of Lytic Mycobacteriophage D29 for the Treatment of Tuberculosis Infection

**DOI:** 10.3390/microorganisms11051214

**Published:** 2023-05-05

**Authors:** Vadim Vadimovich Avdeev, Victor Vladimirovich Kuzin, Mikhail Aleksandrovich Vladimirsky, Irina Anatol’evna Vasilieva

**Affiliations:** National Medical Research Center of Phthisiopulmonology and Infectious Diseases of the Health Ministry of the Russian Federation, Moscow 103030, Russia; vadim.avdeev@rambler.ru (V.V.A.); kuzin@obolensk.org (V.V.K.);

**Keywords:** tuberculosis treatment, lytic mycobacteriophages, liposomes, TB granuloma, experiments in vivo

## Abstract

We have studied the antimycobacterial efficacy of the liposomal preparation of mycobacteriophage D29 on models of tuberculous granuloma in vitro and in the experiment on laboratory mice of the relatively resistant strain C57BL/6, infected with the virulent strain of *M. tuberculosis* H37Rv. We have shown the preparation of liposomal preparation of the lytic mycobacteriophages and its characteristics. The experiments showed a significant lytic effect of the liposomal form of mycobacteriophage D29 both on the model of tuberculous granuloma formed by human blood mononuclear cells in vitro, which is formed in the presence of *Mycobacterium tuberculosis* and on the model of tuberculous infection in C57BL/6 mice. Keywords: mycobacteriophage D29, *M*. *tuberculosis*, liposomes, tuberculous granuloma in vitro, tuberculosis infection and its treatment.

## 1. Introduction

Tuberculosis (TB) remains one of the most common and dangerous infectious diseases and is included in the list of socially significant diseases and of diseases that pose a danger to others. Despite the general improvement in the TB epidemic situation in the world, the problem of the spread of drug-resistant forms of the pathogen—*Mycobacterium tuberculosis* (MTB) is extremely urgent [1]. WHO estimates that only half of treated MDR patients were successfully treated [2]. Multidrug resistance (MDR) is increasing annually by more than 20% [3]. The relative duration of chemotherapy for TB infection and the known relative toxicity of chemotherapy drugs leads to a decrease in patient adherence to treatment, which, in turn, contributes to the spread of drug-resistant MTB strains. The search for alternative ways to treat TB infection, including drug-resistant forms, may become one of the promising areas in the fight against TB.

The use of lytic mycobacteriophages for the treatment of TB infection in conditions of acquiring resistance to TB drugs is of considerable interest as a prospect for developing an alternative method for treating TB infection [4,5,6,7,8,9,10,11,12,13,14], including infection with drug-resistant mycobacteria [8], as well as against the drug-resistant strain of *M. abscessus* [7]. It should be taken into account that there is only one well-studied strain of lytic mycobacteriophages, namely strain D29. Mycobacteriophage D29 is a specific mycobacterial virus that has unique lytic properties against MTB since its genomic organization does not allow it to transform into a lysogenic or moderate phage form [15].

Mycobacteriophage D29 has also been successfully used for the rapid determination of drug susceptibility or resistance of MTB clinical strains to anti-TB antibiotics [16] When studying the prospects for the use of mycobacteriophages for the treatment of TB, the possibility of their selective transport to granulomatous foci of inflammation is considered and confirmed [7,17]. In this regard, liposomal forms of lytic mycobacteriophage are considered with great interest [18,19,20,21,22]. In our work, we studied the bactericidal effect of the liposomal form of mycobacteriophage D29 with different liposome diameters on the model of TB granuloma in vitro and in vivo on laboratory animals infected with the MTB H37Rv virulent strain.

## 2. Materials and Methods

### 2.1. Production of Mycobacteriophage D29

To obtain the required volume of mycobacteriophage, a suspension of *M. smegmatis* was prepared on a solid nutrient medium (Middlebrook 7H10). This suspension with a minimum density was transferred to a liquid nutrient medium (Middlebrook 7H9) in 0.1 M Tris-HCl pH 7.5 containing 1 mM CaCl_2_. It was incubated for a day; then 2 mL of mycobacteriophage was added with a concentration of at least 10^8^ plaque-forming units (PFU)/mL and incubated for another day on a shaker (3–5 shakes per minute, 37 °C). The resulting suspension was centrifuged at 3800 rpm for 15 min. The supernatant was sterilized by filtration using 0.22 μm filters (Millex GP 0.22 m Millipore). The titer of mycobacteriophage was determined by tenfold titration and inoculation on a solid nutrient medium 7H10.

### 2.2. Development and Determination of Biological Activity of Lytic Mycobacteriophage D29

*M. smegmatis* culture was used to propagate the mycobacteriophage. In total, 100 μL of this culture previously diluted in the liquid nutrient medium Middlebrook 7H9 + 1 mM CaCl_2_ was applied to Petri dishes with a solid nutrient medium and incubated at 37 °C for 15 min. Then, 900 µL of 0.1 M NaCl Tris-HCl-MgSO_4_ phage buffer solution pH 7.5 was added to each pre-prepared test tube. Then, 100 µL of the studied lytic mycobacteriophage was added to the first test tube with a phage buffer solution, after which the mycobacteriophage was serially diluted tenfold up to the tenth test tube. The resulting suspension was added from each dilution to Petri dishes with a culture of *M. smegmatis*. The next day, the calculation of the biological activity of the phage was carried out by multiplying the number of obtained plaques (lysis zones) on plates with a lawn of *M. smegmatis* by the appropriate dilution.

### 2.3. Obtaining a Concentrated Highly Purified Preparation of Mycobacteriophage D29

The obtained mycobacteriophage after propagation in the culture of *M. smegmatis* and determination of its biological activity was purified from the lysate of mycobacteria containing a combination of lipases, proteases, and destroyed DNA molecules using ion-exchange chromatography. We used the column volume of 150 mL with a weakly anionic sorbent Macro-Prep DEAE Media (BioRad) on a BIO-RAD BioLogic LP chromatograph [23]. The phage preparation, on average 50–70 mL, was added to the sorbent column suspended in a buffer solution of 0.1 M NaCl. After obtaining the first peak, the molarity of the solution was increased to 1M NaCl and then the second peak of the phage preparation was collected (Figure 1). The mycobacteriophage preparation was dialyzed against 0.1 M phage buffer solution for 24 h on a magnetic stirrer (110 rpm, 4 °C) and sterilized by filtration through 0.22 µ filters.

### 2.4. Obtaining a Liposomal Preparation of Mycobacteriophage D29 by Extrusion with a Liposome Diameter of 0.45 and 0.8 Microns

To obtain a phospholipid film, 40 mg of phosphatidylcholine, 98% purity, (Sigma-Aldrich, Taufkirchen, Germany) and 13 mg of cholesterol (Sigma Aldrich, Taufkirchen, Germany) were used. All components were dissolved in 3 mL of 95% ethanol; then dried on a rotary evaporator at t 37 °C until a lipid film was obtained. After complete drying, the phospholipid film was removed by adding 3 mL of a purified mycobacteriophage preparation with a high amount of PFU (at least 10^8^/mL). The flask with the phospholipid film was shaken for 5 min. The obtained multilayer liposomes were subjected to extrusion through filters with a pore size of 5 microns once. This was followed by 19 times extrusion through Millipore filters with a pore size of 0.45 or 0.8 microns, respectively (Millex GP), and an additional single extrusion through a filter with a pore size of 0.45 and 0. 8 microns.

### 2.5. Obtaining a Liposomal Preparation of Mycobacteriophage Based on Chromatography on a Sephadex G-75 Column

Sephadex G-75 (Pharmacia Fine Chemicals) was weighed in a phage buffer solution. The size of the weighted Sephadex column is 17 × 1.1 cm. In a 100 mL round-bottom flask, 30 mg of 98% pure phosphatidylcholine and 10 mg of cholesterol were dissolved in 3 mL of 96% ethanol solution. The lipid film was prepared using a rotary evaporator, which was removed by shaking using 3 mL of a phage buffer solution to obtain a micellar suspension of multilayer-preparation liposomes. To obtain single-layer liposomes, 300 μL of a 20% solution of sodium deoxycholate in 20% ethanol was added to the resulting suspension. A four-fold volume of the phage preparation compared to the micellar solution was preliminary added to the Sephadex column, and chromatography was stopped. Then, 3 mL of micellar solution with sodium deoxycholate was added, and very slow chromatography was performed at a rate of 0.6 mL/min. Opalescent fractions were collected at the outlet. Then, 8 mL of the most opalescent preparation was obtained, an aliquot of which was precipitated only by centrifugation at 14.5 thousand rpm.

### 2.6. Obtaining Tuberculous Granuloma In Vitro, Formed by Human Blood Mononuclear Cells

To obtain a granuloma, the venous blood of a healthy adult volunteer with a positive reaction during antigen-specific induction of interferon-gamma in a volume of 20 mL was used. In test tubes with a volume of 15 mL, 3 mL of a solution of the Ficoll-Paque (“GE Healthcare”) preparation was poured, on which 4 mL of diluted 1:1 saline blood was layered. It was then centrifuged at 2000 rpm for 20 min. Then, the cellular interphase formed at the media interface was collected in a 50 mL tube. Washed two times—the first time with physiological saline (30 mL); the second time with RPMI 1640 medium without additives (20 mL). Centrifuged at 3000 g for 15 min, the supernatant was removed. The number of cells was counted using a Scepter cell counter. The resulting suspension of mononuclear cells was adjusted to a concentration of at least 5.0 × 10^5^/mL and poured into a 24-well plate in 500 µL of the suspension. A total of 500 μL of RPMI 1640 medium with 10% FBS was added to each well of the plate. A composition of RPMI 1640 culture medium with 10% FBS: (1) RPMI 1640–500 mL; (2) FBS–50 mL; (3) Pen Strep 100×–5 mL; (4) Alpha-Glutamine–5 mL; (5) MEM Vitamins solution–5 mL; 100×–5 mL; (6) MEM NEAA–5 mL; (7) Sodium Pyruvate 100×–5 mL.

The next day, 50 µL of H37Rv 10^5^/mL was added to each well of the plate. The medium was changed every two days. The cell culture was monitored using an inverted Leica DMIL microscope at a magnification of 1:200 with a Leica DFC 420 digital camera. Granuloma formation occurred on the 13–15th day. (Figure 2).

## 3. Results and Discussion

### 3.1. Manufacturing a Liposomal Preparation of Mycobacteriophage D29

To obtain a higher concentration of mycobacteriophage after chromatography and dialysis, the drug was centrifuged in centrifuge concentrators at 1500 rpm for 15 min. The percentage of mycobacteriophage incorporation into liposomes was evaluated by quantitative determination of mycobacteriophage DNA during real-time PCR (RT-PCR) [16]. The percentage of mycobacteriophage incorporation into liposomes was at least 25% in different experiments (Table 1). When using the extrusion method with filters of 0.8 microns, the level of inclusion of mycobacteriophage was 40%; see Table 1 and electron microscopy (Figure 3).

When conducting electron microscopy, the size of the resulting liposomes obtained by column chromatography was 250 nm (Figure 4). The resulting sterile solution of phage liposomes was stored at 4 °C. Mycobacteriophage not included in liposomes was collected and concentrated for further use. For quantification of incorporation of the mycobacteriophages into liposomes, their samples (in a volume of 200 µL) were centrifuged at 14 thousand rpm (refrigerated centrifuge, Thermo Electron, Waltham, MA, USA), and the sediment volume was measured. The amount of mycobacteriophage DNA in the starting material and in the sediment was determined using real-time PCR (“Amplitub RT, Synthol, Moscow, Russia). In this case, the inclusion percentage was calculated based on the differences between the threshold cycles (the number of threshold cycles is inversely proportional to the amount of DNA in the samples).

### 3.2. Formation of a Human Granuloma by Blood Mononuclear Cells In Vitro and Results Analysis Antimycobacterial Efficiency Liposomal

After the formation of a granuloma, mycobacteriophage and liposomal preparations of mycobacteriophage D29, obtained via various methods with the activity determined by real-time PCR, were added to the wells of the plate in a volume of 100 μL. A day later, the procedure was repeated. Two days later, the contents of the wells of the plate were shown on a solid nutrient medium.

In the experiments performed on the in vitro TB granuloma model formed by human blood mononuclear cells, using a liposomal mycobacteriophage obtained by extrusion, a significant antimycobacterial effect was established, obviously exceeding the antimycobacterial effect of mycobacteriophage not included in liposomes (Figure 5). In the case of positive control, more than ten large MTB colonies were detected in the plate. Growth of MTB was not detected for liposomal preparation of mycobacteriophage D29. Non-encapsulated (free) preparation of mycobacteriophage D29 produced more than ten small MTB colonies.

### 3.3. Studies of the Antimycobacterial Activity of the Liposomal Preparation of Mycobacteriophage D29 on Laboratory Mice of the C57BL/6 Strain

The study was conducted on the basis of the Scientific Center for Applied Microbiology (Obolensk, Moscow Region) in accordance with the veterinary protocol N 634- n\22, 15 August 2022, of the bioethics commission, “Modeling of tuberculosis infection, which provides for the humane treatment of animals s C57BL/6 strain”.

To determine the in vivo activity of the liposomal form of the lytic mycobacteriophage D29, an experiment was carried out on laboratory mice of the genetically resistant strain C57BL/6. Laboratory mice were infected with an MTB H37Rv virulent strain. Infection of laboratory mice was carried out by intranasal (on inspiration) administration of 50 µL of a suspension of MTB at a concentration of 2.10^5^ bacilli per ml. Infected mice were divided into three groups of six mice with different treatment regimens starting from day 35 after infection. In the first group, lytic mycobacteriophage D29 was administered in liposomal form, obtained by extrusion, in a volume of 50 μL/animal, intranasally (on inspiration), two times a day, three times with an interval of one day. The second group of mice was injected intranasally with a liposomal preparation of lytic mycobacteriophage D29 obtained by the column method in the same volume and mode of administration. In the third group—control, infected mice were not treated. Mice were euthanized 5 days after treatment. The results were evaluated (a) by macroscopic evaluation of cured lung preparations; (b) when inoculating the lungs on plates with Middlebrook 7H10 nutrient medium; (c) according to the results of histological examination of fixed lung preparations.

The results of evaluating the antimycobacterial effect of two liposomal forms of mycobacteriophage D29 on laboratory mice C57BL/6 infected intranasally are presented on Table 2.

Certainty that the liposomal form of lytic mycobacteriophage D29 shows a significant antimycobacterial effect. At the same time, the preparation of liposomal mycobacteriophage obtained by extrusion proved to be more effective than the preparation obtained by the column method. It is possible that this effect is associated with a slightly larger size of liposomes (0.8 microns), which provided a larger amount of lytic mycobacteriophage included. In general, the results of the presented studies showed a positive outlook for the studied liposomal preparations, which allows us to count on their effectiveness after appropriate biosafety studies and in clinical studies, possibly with endobronchial administration in patients with pulmonary TB.

Histological detection among the lungs in the control group of infected animals (3) showed signs of diffuse inflammation (Figure 6A); in Group (2), the lungs of mice that received a liposomal preparation obtained by the column chromatography method detect foci of diffuse inflammation and fibrotic changes in the lung tissue (Figure 6A,B), while in Group (1), the lungs of mice that received the preparation obtained by extrusion with a high level of increase in mycobacteriophage included in liposomes lung tissue remained intact. (Figure 7).

## 4. Conclusions and Short Discussion

Thus, we can state that the use of the liposomal form of lytic mycobacteriophages D29 with a liposome size of 0.8 microns showed the most pronounced significant antimycobacterial effect.

We believe that in this work, for the first time on an experimental model of laboratory animals with developed tuberculosis infection, the high efficiency of using the liposomal form of lytic mycobacteriophages in short–term (three days) endobronchial (endonasal on inspiration) administration was proved.Possible questions about the toxicity of such a drug are planned to be resolved as part of the forthcoming studies on the acute and chronic toxicity of our drug on laboratory animals in the near future. We believe that the success of treatment described in the well-known study using a combination of lytic mycobacteriophages to treat a lung transplant girl infected with drug-resistant *M. abscessus* infection [7] suggests minimal toxicity of specific lytic mycobacteriophages in their clinical use.We believe that a liposomal preparation of lytic mycobacteriophages in the future can be administered to patients with tuberculosis into the bronchial system through a bronchoscope. However, the possibilities of inhalation administration are also discussed [24].It also planned to conduct a comparative study of the effectiveness of liposomal preparation of the lytic mycobacteriophages and a combination of the most well-known and effective antibiotics for the treatment of tuberculosis.

## Figures and Tables

**Figure 1 microorganisms-11-01214-f001:**
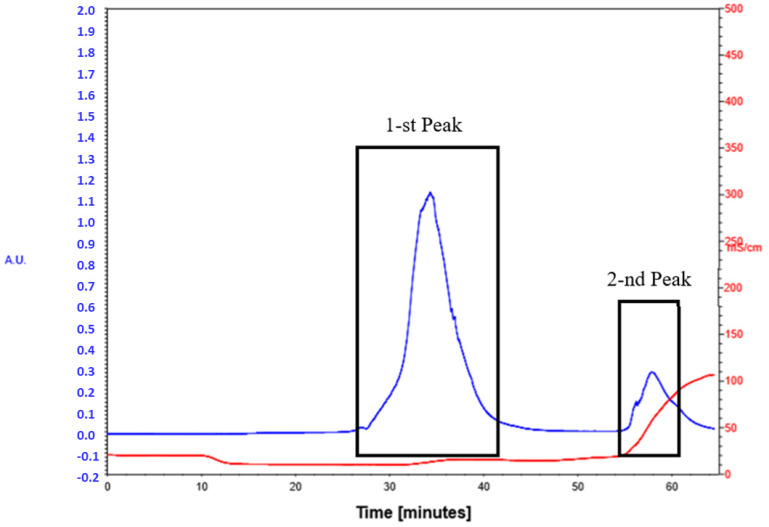
Chromatography peak image of lytic mycobacteriophage D29. 

 Optical density [u] 

 Conductivity [mS/cm].

**Figure 2 microorganisms-11-01214-f002:**
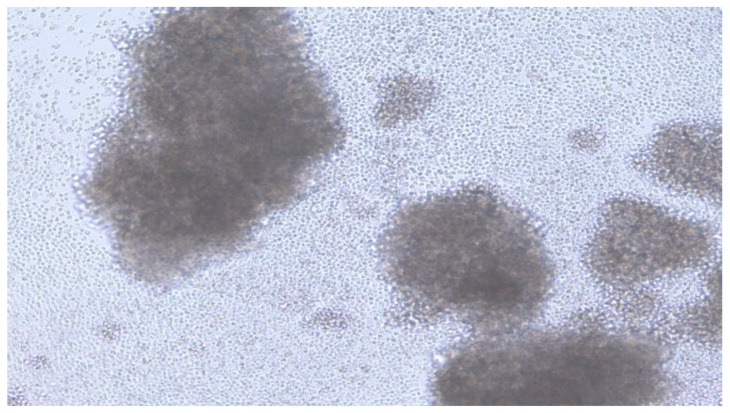
Image of TB granuloma on the day 14.

**Figure 3 microorganisms-11-01214-f003:**
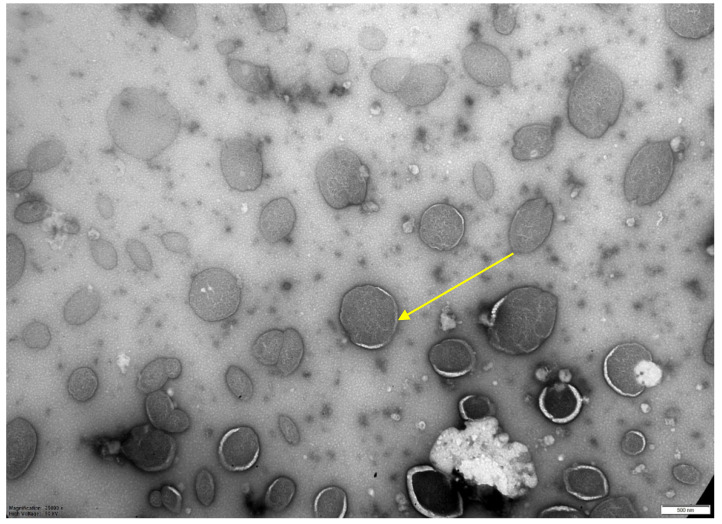
Electron microscopy of a liposomal preparation of mycobacteriophage D29 obtained by extrusion (liposome diameter 0.8 microns). Magnification 1:25,000. Scale—500 nm.

**Figure 4 microorganisms-11-01214-f004:**
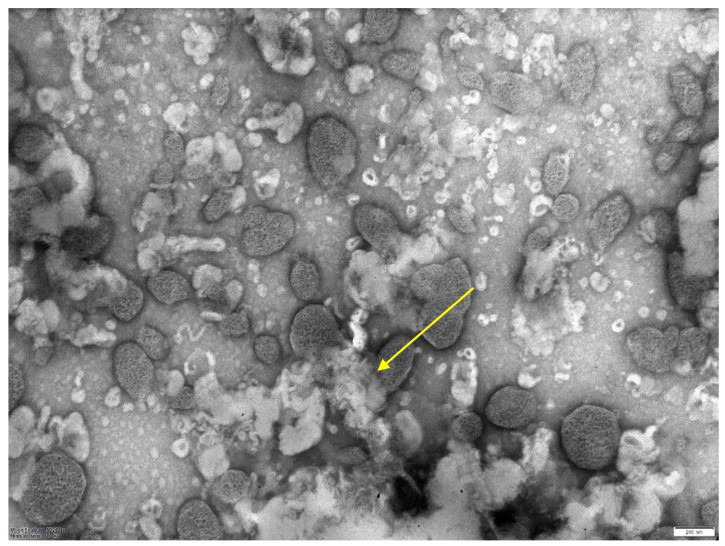
Electron microscopy of a liposomal preparation of mycobacteriophage D29 obtained by column chromatography. Scale—200 nm. The average size of liposomes is 0.25 microns.

**Figure 5 microorganisms-11-01214-f005:**
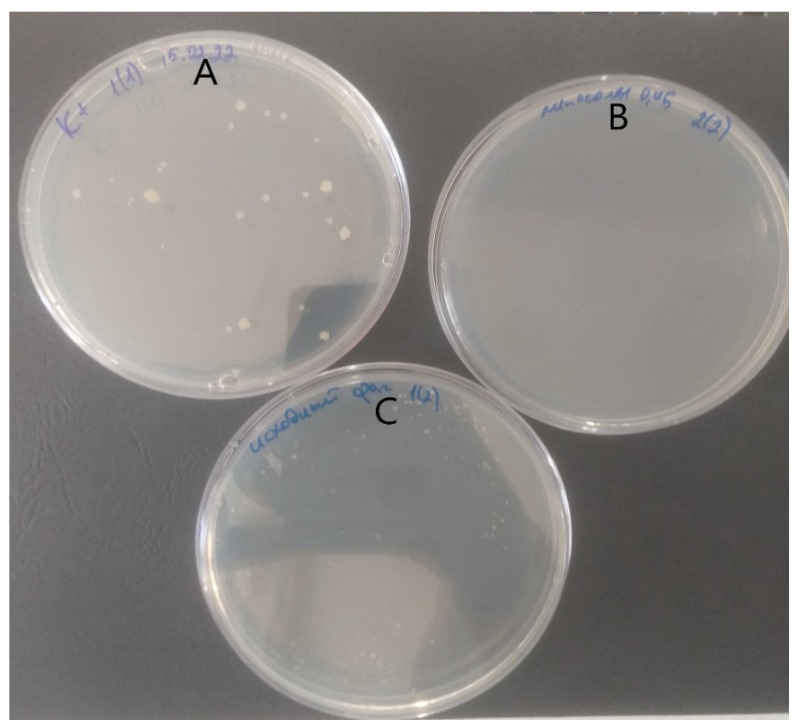
Comparison of culture results obtained in an in vitro TB granuloma experiment formed by human blood mononuclear cells. (**A**) positive control, (**B**) liposomal preparation of mycobacteriophage D29, and (**C**) non-encapsulated (free) preparation of mycobacteriophage D29.

**Figure 6 microorganisms-11-01214-f006:**
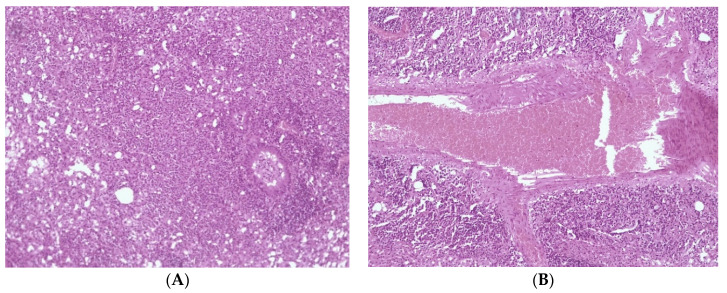
Histological picture of changes in the lung tissue of the mouse. (**A**) diffuse inflammation of the lung tissue, (**B**) perivascular fibrosis, and plethora.

**Figure 7 microorganisms-11-01214-f007:**
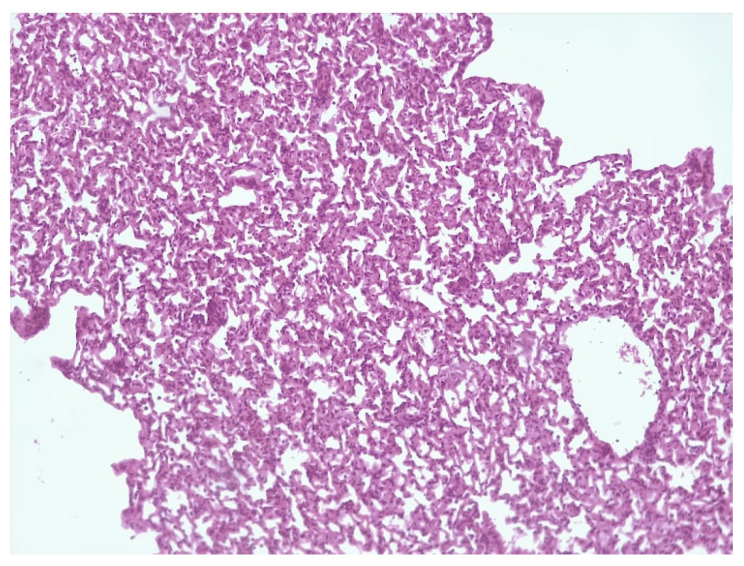
Intact lung tissue.

**Table 1 microorganisms-11-01214-t001:** Efficiency of incorporation of mycobacteriophage D29 into liposomal preparations.

Study Drug	Threshold Cycle for the Supernatant	Threshold Cycle for the FAM Sediment	Percentage of Mycobacteriophage Incorporation into Liposomes
Parent lytic mycobacteriophage D29	16.75	*	
Liposomal mycobacteriophage D29, 0.8 micron (extrusion)	17.25	15.27	40
Liposomal mycobacteriophage D29, 0.25 micron (chromatography)	18.21	15.56	32
Liposomal mycobacteriophage D29, less than 0.45 micron (extrusion)	17.14	15.12	25

* Mycobacteriophage not included in liposomes is deposited only during ultracentrifugation, at least 40 thousand rpm.

**Table 2 microorganisms-11-01214-t002:** Efficacy results of liposomal preparations of lytic mycobacteriophage in macroscopic evaluation and microbiological examination of lung organs of C57BL/6 mice, infected mycobacteria of the virulent strain H37Rv.

Group Number/Mouse No.	Treatment Regimen	Mouse Weight Gr.	Lung Weight Gr.	Macroscopic Assessment of Lung Condition	Culture Results on a Middlebrook 7H10 Solid Medium (CFU)
1/1	Liposomal form of the lytic mycobacteriophage D29 obtained by extrusion method	22.2	0.24	normal	No growth
1/2		26.6	0.33	Foci hyperemia	3 · 10 lg
1/3		25.5	0.32	normal	No growth
1/4		25.5	0.23	normal	No growth
1/5		21.8	0.38	Foci hyperemia	3 · 10 lg
1/6		21.9	0.21	normal	No growth
2/1	Liposomal form of the lytic mycobacteriophage D29 obtained by chromatography column method	24.3	0.24	normal	No growth
2/2		22.3	0.26	normal	Singles colonies
2/3		18.1	0.64	Multiple lesions	4 · 10 lg
2/4		22.4	0.36	Multiple lesions	4 · 10 lg
2/5		23.8	0.32	normal	No growth
2/6		24.5	0.50	Multiple lesions	5 · 10 lg
3/1	Control. MTB H37Rv without treatment	23.7	0.35	Single Foci lesions	5 · 10 lg
3/2		24.4	0.30	Single Foci lesions	4 · 10 lg
3/3		22.9	0.73	Multiple lesions	5 · 10 lg
3/4		24.5	0.70	Multiple lesions	Singles colonies
3/5		23.2	0.59	Multiple lesions	5 · 10 lg

## Data Availability

Not applicable.

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
