# Peer review of "Experimental Studies of the Liposomal Form of Lytic Mycobacteriophage D29 for the Treatment of Tuberculosis Infection"

_microorganisms, 2023, doi:10.3390/microorganisms11051214_

Round 1

Reviewer 1 Report

The authors present in this manuscript studies on the use of lytic mcyobacteriophage D29 for the potential treatment of tuberculosis. The lysate was used both on a model of tuberculous granuloma formed by human blood mononuclear cells in vitro and on a model of tuberculous infection in C57BL/6 mice. The design of the experiments makes sense and the results are quite interesting. However, the manuscript needs much improvement to achieve the standards of this Journal. The language has to be corrected, sentences need to be cut into smaller, more concise phrases and syntax must be checked. There is some repetition between "materials and methods" and "results and discussion". Author contributions could be added. I believe that the authors wrote the manuscript without paying much attention to detail and this should be corrected before this manuscript is considered for publication. The authors need to revisit the text to make it more comprehensive and explanatory.

Author Response

Dear colleague! Thank you for reviewing our article!

We have taken into account your important comments. In particular, we removed the erroneous repetitions of the text in the sections “materials and methods” and “research results”. 

Unfortunately, we are not native English speakers, but it seems to us that this should not significantly interfere with understanding the content of our work.

Reviewer 2 Report

General comment:

The authors encapsulated mycobacteriophage D29 into liposome to efficiently kill Mycobacterium tuberculosis parasite in human cell. The idea is basically good and relevant to attack M. tuberculosis in phagocytes, but this manuscript has some critical problems. The authors may need to improve the manuscript accordingly.

Specific comments:

1.       Whole text: It will be not a responsibility of the authors but the editorial office that the manuscript is already in publication form then is very much inconvenient for reviewing. Number for each line should be for easy review.

2.       Introduction: Reference #18 is inappropriate to support this sentence. The authors should refer the original article.

3.       Materials and methods: 2.2 CaCl2 -> CaCl2

4.       Materials and methods: There is no methodology description about animal experiment in this section.

5.       Results and Discussion: The lytic mycobacteriophage 29 -> The lytic mycobacteriophage D29

6.       Results and Discussion: There are unnecessary repetitions in this section; “The lytic mycobacteriophage 29 obtained after propagation in the culture of M. smegmatis was purified from the mycobacterial lysate by the ion-exchange chromatography. The phage preparation, on average 50–70 ml, was added to the sorbent column suspended in a buffer solution of 0.1 M NaCl. After obtaining the first peak, the molarity of the solution was increased to 1M NaCl and then the second peak of the phage preparation was collected (Figure 1)” ” To obtain a granuloma, venous blood of an employee with a positive reaction during antigen-specific induction of interferon gamma in a volume of 20 ml was used.”

7.       Results and Discussion: What is the reason for the colony number and size difference in between dish A and dish C? The number of colonies growing on dish A is relatively low and it may affect the reliability of the experiment. It is also recommended to include D29 free liposome treatment control.

8.       Tables: The decimal point will usually be a period rather than a comma. In addition, I don't understand what the CFU notation means in Table 2. Please use the standard notation method.

9.       IRB statement: Please include animal ethics statement.

Author Response

Dear colleague! Thank you for reviewing our article and for the fair and important remarks made.

We have removed link number 18 and fixed random errors! We  removed the erroneous  repetitions  of the text in the sections “materials and methods” and “research results”.    The methodology for working with animals in the framework of the experiment is described in research results section. We also fixed bugs in numbers: dots instead of commas. Regarding the remark on the number of MTB colonies on plats: A and C Fig.№ 5 we know that one large colony corresponds to at least 10 small MTB colonies. In the section on animal experiments, we have included information with relevant protocol on the bioethics of working with animals.

Reviewer 3 Report

This study evaluated the anti-mycobacterial effect of Mycobacteriophage D29 on in vitro tuberculosis granuloma model and experimental mouse model, and found that liposomal-type Mycobacteriophage D29 showed significant lytic effect on both tuberculosis granuloma model and mouse tuberculosis infection model. Granulomas play an important role in the pathogenesis and damage of Mycobacterium tuberculosis. In-depth research on granulomas will help to understand the mechanism of action between Mycobacterium tuberculosis and the host, and provide new insights for the prevention and treatment of tuberculosis. Therefore, this study is somewhat innovative in this respect. However, some issues should be amended before it was accepted for publication.

1.        Abstract: a more detailed description is needed in the Abstract section to give enough information to the readers. The number of key words should be more than 5.

2.        Materials and Methods Section: The manufacturer, model and country of some key reagents or instruments should be provided to ensure the reproducibility of all experiments.

3.        Results and discussion: (1) Figure 1: Some words and edge of the box seems missing, please carefully check it; (2) The English language should be improved by an English native speaker, such as “To obtain a liposomal preparation of mycobacteriophage D29 by extrusion the flask with the phospholipid film and mycobacteriophage preparation was shaken for 5 minutes ……”; (3) The scale notes in Figure 2 and 3 are too small to read clearly; (4) Liposomal preparation of mycobacteriophage D29 should be indicated in Figure 2 and 3.

4.        From Figures 2 and 3, we can find that the size of the liposomal preparation of mycobacteriophage D29 is not uniform, and the difference is large, indicating that the effect of this method is not stable. The authors should explain this issue.

5.        Figure 4, Why did the authors not use HE staining and immunohistochemistry to analyze the granulomas? The cell type of granulomas should also be analyzed with methods such as fluorescent labeling.

6.        The authors' analysis of pathology was too simple to support the conclusions drawn in this study. It is suggested that the author should do more in-depth research on the pathological part, such as immunohistochemistry, multi-color fluorescence labeling, immune electron microscopy and other techniques.

7.        In the section of Results and Discussion, authors only showed their results but failed to give a good discussion.

8.        The format of the article should be strictly adjusted according to the template of the journal, such as font, size, etc.

Author Response

 Dear colleague! Thank you for reviewing our article and for the fair and important remarks made.!

We have expanded   the abstract of the article somewhat and included keywords. We removed duplicates in the Materials and Methods and results sections.

We have inserted frames in Fig.1. We have added data on manufactures of reagents and equipments. On fig 2 and 3 by electron microscopy indicated the scale of the figures  and arrows for liposomes. We have removed the duplicates in the “methods” and “results”.

As for the methods of immunohistochemistry of TB granulema, it seems to us that is a separate subject of research for characterizing the TB granuloma itself. We are willing to do this as a part of separate important and interesting project to investigate the mechanisms of immunological response and resistance to TB infection, if we receive appropriate funding.

Once again ,thank you for reviewing our work.

Round 2

Reviewer 1 Report

The authors revised the manuscript according to most of my comments so I suggest acceptance under minor revision - if possible and if agreed by the editor - in text editing. Even if there is no text editing, I believe that the manuscript in its present form is comprehensive.

Author Response

We are grateful to the Reviewer for the positive evaluation of our work. Minor text editing has been done. 

Reviewer 3 Report

The authors have addressed all of my concerns, and I would like to suggest acceptance.

Author Response

We are grateful to the Reviewer for the positive evaluation of our work.